# Prevalence of clinically manifested drug interactions in hospitalized patients: A systematic review and meta-analysis

**Tâmara Natasha Gonzaga de Andrade Santos**[1☯], **Givalda Mendonça da Cruz Macieira**[1☯], **Bárbara Manuella Cardoso Sodré Alves**[1☯], **Thelma Onozato**[1☯], **Geovanna Cunha Cardoso**[1☯], **Mônica Thaís Ferreira Nascimento**[1☯], **Paulo Ricardo Saquete Martins-Filho**[2☯], **Divaldo Pereira de Lyra, Jr.**[1☯], **Alfredo Dias de Oliveira Filho**[1☯]*

1 Department of Pharmacy, Laboratory of Teaching and Research in Social Pharmacy (LEPFS), Federal University of Sergipe, São Cristóvão, Sergipe, Brazil, 2 Universitary Hospital, Investigative Pathology Laboratory, Federal University of Sergipe, Aracaju, Brazil

☯ These authors contributed equally to this work.
* adias@hotmail.com

## Abstract

### Aims

This review aims to determine the prevalence of clinically manifested drug-drug interactions (DDIs) in hospitalized patients.

### Methods

PubMed, Scopus, Embase, Web of Science, and Lilacs databases were used to identify articles published before June 2019 that met specific inclusion criteria. The search strategy was developed using both controlled and uncontrolled vocabulary related to the following domains: "drug interactions," "clinically relevant," and "hospital." In this review, we discuss original observational studies that detected DDIs in the hospital setting, studies that provided enough data to allow us to calculate the prevalence of clinically manifested DDIs, and studies that described the drugs prescribed or provided DDI adverse reaction reports, published in either English, Portuguese, or Spanish.

### Results

From the initial 5,999 articles identified, 10 met the inclusion criteria. The pooled prevalence of clinically manifested DDIs was 9.2% (CI 95% 4.0–19.7). The mean number of medications per patient reported in six studies ranged from 4.0 to 9.0, with an overall average of 5.47 ± 1.77 drugs per patient. The quality of the included studies was moderate. The main methods used to identify clinically manifested DDIs were evaluating medical records and ward visits (n = 7). Micromedex® (27.7%) and Lexi-Comp® (27.7%) online reference databases were commonly used to detect DDIs and none of the studies evaluated used more than one database for this purpose.

**Data Availability Statement:** All relevant data are within the manuscript and its Supporting Information files.

**Funding:** This study was financed by the Coordenação de Aperfeiçoamento de Pessoal de Nível Superior - Brasil (CAPES) - Finance Code 001. There was no additional external funding received for this study. The funder had no role in study design, data collection and analysis, decision to publish, or preparation of the manuscript.

**Competing interests:** The authors have declared that no competing interests exist.

## Conclusions

This systematic review showed that, despite the significant prevalence of potential DDIs reported in the literature, less than one in ten patients were exposed to a clinically manifested drug interaction. The use of causality tools to identify clinically manifested DDIs as well as clinical adoption of DDI lists based on actual adverse outcomes that can be identified through the implementation of real DDI notification systems is recommended to reduce the incidence of alert fatigue, enhance decision-making for DDI prevention or resolution, and, consequently, contribute to patient safety.

## Introduction

Medicines play an important role in the prevention of diseases and the promotion, maintenance, and recovery of a patient's health, thereby contributing to improvements in the quality of life and life expectancy of the population [1–3]. Despite these benefits, problems with pharmacotherapy are becoming more frequent and occur in 42–81% of hospitalized patients [4–7]. These complications, defined as events or circumstances involving pharmacotherapy that actually or potentially interfere with the desired health outcome [8], include inadequate medication or dosage, adverse reactions, and drug-drug interactions (DDIs) [9].

A DDI, defined as a change in the effect of a drug as a result of the interaction with one or more drugs, may cause a reduction or an increase in therapeutic efficacy [10,11]. Undesirable DDIs are a major health concern, particularly in the hospital setting. Hospitalized patients generally have polypharmacy and complex pharmacotherapy, which, together with clinical instability, may result in adverse outcomes, such as clinical deterioration and increased length of hospitalization, but may also lead to death [12]. A study of hospitalized patients revealed that the DDIs between warfarin-aspirin and digoxin-atenolol were associated with primary intracerebral hemorrhage and cardiac rhythm disorders, respectively [13]. In a recent study, a recurrent clinically manifested DDI of methyldopa with ferrous sulfate, in which one drug made the other less effective, resulted in an increase in systolic blood pressure (BP) in all high-risk pregnant women who were evaluated. After ferrous sulfate was discontinued, a reduction was noted in the BP levels of patients [14].

Several databases have been developed to assist prescribers in the identification of DDIs [15]. As these databases contain a large number of DDIs, there may be excessive and nonspecific alerts that lack focus on the clinical relevance and correct management of DDIs [16]. The excessive number of unconfirmed warnings of clinical manifestations has led to an effect known as "alert fatigue," which is a condition wherein prescribers ignore relevant alerts when receiving many notifications [17]. Recent studies have shown that 69–91% of DDI alerts communicated to prescribers were ignored because the DDIs were not considered to be manifested [18–20].

Most studies on this subject do not focus on the prevalence of DDIs that manifest clinically [21,22]. A systematic review and meta-analysis of the harmful effects of DDIs in hospitalized patients did not focus on clinically manifested DDIs. This review included studies that investigated only potential and/or clinically relevant DDIs, and studies with sufficient data to allow independent readers to calculate the prevalence of clinically manifested DDIs fully available were not actively searched [21]. Thus, the present systematic review and meta-analysis aimed to determine the prevalence of clinically manifested DDIs in hospitalized patients.

## Methods

This systematic review and meta-analysis were carried out following the MOOSE (Meta-analysis of Observational Studies in Epidemiology) statement [23]. The protocol for this study was registered in the PROSPERO international prospective register of systematic reviews database (CRD 42017056856).

### Search question

To clarify our hypothesis, eligibility criteria, and search strategy, we used the PICO elements (P: hospitalized patients; I: Drug-Drug Interactions; C: not applied; O: clinically manifested DDIs) [24] to formulate the following research question: which one the prevalence of clinically manifested DDIs in hospitalized patients?

### Data source and search strategy

To determine the prevalence of clinically manifested DDIs in hospitalized patients, a comprehensive literature search was conducted using the PubMed, Scopus, Embase, Web of Science, and Lilacs databases for articles published up to June 2019. Indexed terms from Medical Subject Headings (MeSH) and other search terms for "drug interactions," "clinically relevant," and "hospital" were used to identify the articles. Other term considered was "clinically manifested", dropped due because the terminologies for manifested DDIs used in the retrieved studies were not related to the search term. Each term was grouped through Boolean operators (AND; OR) to their synonyms and sub- categories and adapted to each database. The full search strategies can be found in S1 Table. In this systematic review, clinically manifested DDIs were defined as DDIs with clinical implications, excluding theoretical interactions, even if they were tagged as "clinically relevant" DDIs.

### Study selection

Original observational studies were included if they met the following criteria: *(a)* the identification of DDIs was performed by using a DDI electronic database; *(b)* clinically manifested DDI was confirmed by laboratory tests and/or signs and symptoms were documented in the medical records and analyzed by specialists [25]; *(c)* data for the calculation of the prevalence of clinically manifested DDIs among patients, prescriptions, or DDI adverse reaction reports were available; and *(d)* the study was published in English, Portuguese, or Spanish. In this systematic review, we excluded: *(a)* duplicate records; *(b)* studies with unavailable abstract or full-text, even after authors were contacted; and *(c)* studies focusing only on specific diseases/pharmacotherapies (for example: patients receiving oncological, HIV, or diabetes treatment) or specific drugs.

Two reviewers (B.M.C.S and T.N.G.A) independently selected the studies and manually screened potentially relevant titles, followed by the abstracts, and full texts. After a thorough reading of the selected texts, references from these studies were analyzed in order to identify other potentially relevant studies. Differences between the reviewers' decisions were analyzed and resolved by a third reviewer (G.C.C). The degree of agreement among reviewers was measured by using the Cohen *Kappa* index [26].

### Data extraction

The following information was extracted: author names, year of publication, country, practice setting, sample (type and number of participants), study design, study duration, detection method of manifested drug interactions, database used, severity of drug interactions,

prevalence rate of clinically manifested DDIs, terminology used to address manifested drug interaction, main limitations, and methodology biases.

Two reviewers (T.N.G.A and G.M.C.M) independently extracted the data. Differences were resolved by discussion between the two reviewers.

## Quality assessment

The Newcastle-Ottawa Scale (NOS) was used to assess the quality of the case-control studies [27]. The quality of the cross-sectional and prospective studies was assessed using the "Quality Assessment Tool for Observational Cohort and Cross-Sectional Studies" [28]. This tool measures 14 different criteria which are then used to give each study an overall quality rating of good ($\geq$12), fair (5–11), or poor (<5) [28]. Two reviewers (T.N.G.A., and G.C.C.) independently performed the validity assessment. All discrepancies were resolved by discussion between the two reviewers.

## Statistical analysis

Two-sided confidence intervals for the single proportions were calculated according to Newcombe's method [29]. We performed a meta-analysis of the prevalence of manifested DDI according to practice setting using the logit transformation and a random-effects model. Heterogeneity was assessed using the $I^2$ value [30]. Meta-analysis was conducted in RStudio (version 0.98.1083).

## Results

### Selection of studies

The initial search of the selected databases identified 5,999 studies. Of these, 10 studies (6,541 patients) met the inclusion criteria. The selection process and the number of articles excluded at each stage of this systematic review are presented in Fig 1.

The degree of agreement between the two primary evaluators (B.M.C.S. and T.N.G.A.) was excellent for the screening of titles (k1 = 0.94), moderate for abstracts (k2 = 0.55), and excellent for full texts (k3 = 0.92).

### Characteristics of the studies

The studies included were conducted in Europe (n = 8) [31–38], Asia (n = 1) [39], and North America (n = 1) [40]. The methodological designs of the selected studies were: cross-sectional (n = 4) [34–36,40]; prospective longitudinal (n = 5) [31,33,37–39]; and a single case-control (n = 1) [32]. There were large variations in sample sizes (82–3,473 patients). With regard to the hospital setting, the studies were performed in internal medicine units (n = 5) [31–34,37], emergency units (n = 3) [35–36,40], an intensive care unit (ICU) (n = 1) [39], and a geriatric unit (n = 1) [38] (Table 1).

### Prevalence of clinically manifested DDIs

The prevalence of clinically manifested DDIs reported in individual studies ranged between 1.2% and 64.0% (Table 2). The highest prevalence was reported in the study by Ray et al. (2010), which evaluated the incidence of adverse reactions caused by DDIs in 400 patients admitted to an ICU [39]. The lowest prevalence of clinically manifested DDIs was found in a cross-sectional study conducted by Fokter et al. (2010), which evaluated only medical records to determine DDIs manifestations in 323 patients of an internal medicine ward [34] (Table 1).

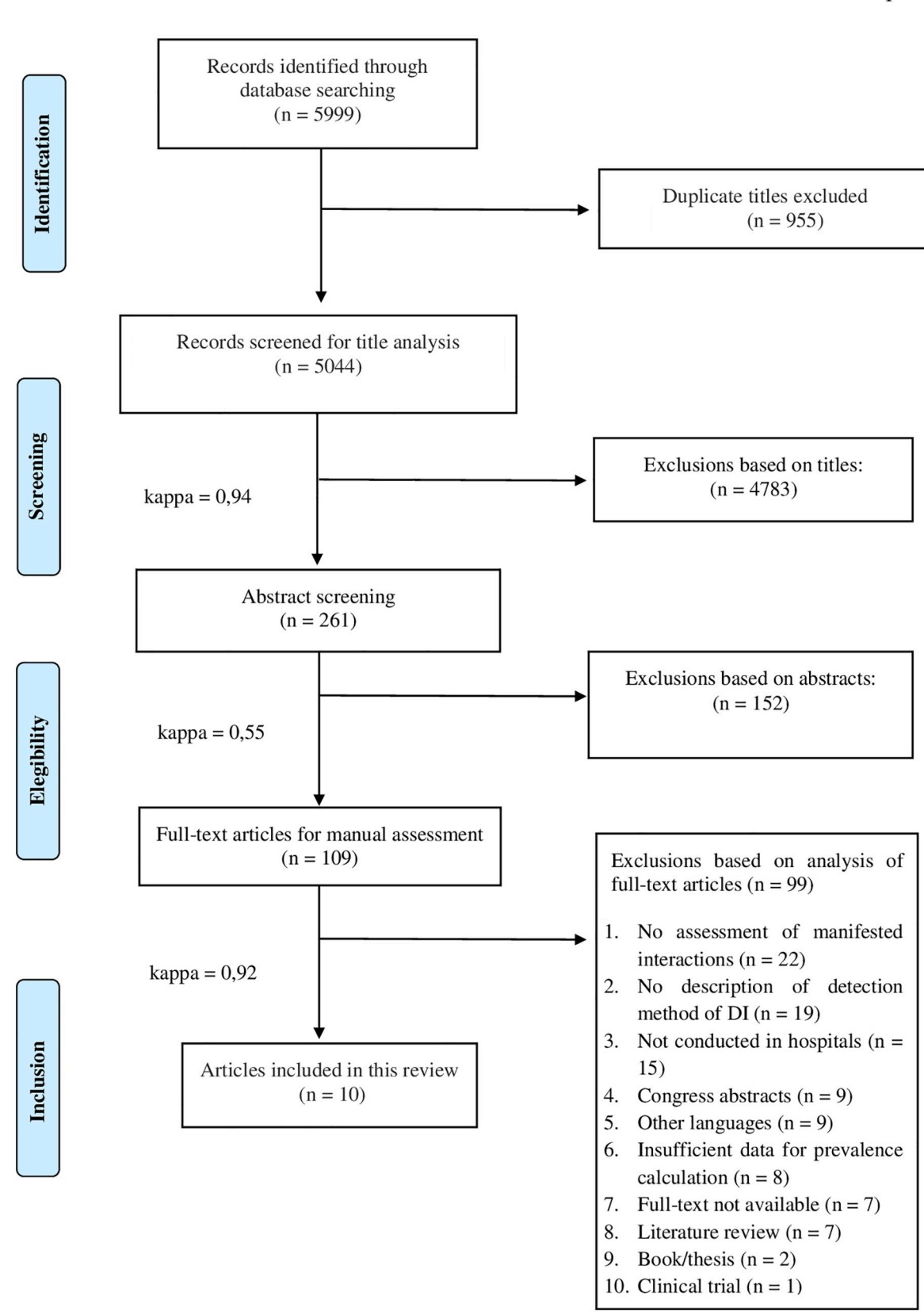

**Fig 1. Flow diagram describing the selection process of the study.**

**Table 1. Characteristics of studies assessing drug interactions in hospitalized patients.**

| Author, year | Country | Study design | Duration | Setting | Detection method of DI | Database | Sample size | Number of clinically manifested DDIs | Main limitations |
|---|---|---|---|---|---|---|---|---|---|
| Herr et al., 1992 | Canada | Cross-sectional | 1 month | Emergency | Medical record and Ward visit | Hansten Drug Interaction Knowledge | 340 patients | 5 | NR |
| Egger et al., 2003 | Germany | Prospective longitudinal | 4 months | Geriatric unit | Medical record and Ward visit | NR | 163 patients | 26 | NR |
| Blix et al., 2008 | Norway | Multicenter prospective | 10 months | Internal medicine | Medical record and Ward visit | Stocley® | 827 patients | 99 | NR |
| Fokter et al., 2009 | Slovenia | Cross-sectional | 12 months | Internal medicine | Medical record | Micromedex® | 323 patients | NR | Retrospective study; Sample size |
| Ray et al., 2010 | India | Prospective longitudinal | 10 months | Intensive care unit | Medical record and Interview | Epocrates® | 400 patients | 208 | NR |
| Muñoz-Torrero et al., 2010 | Spain | Case control | 2.5 months | Internal medicine | Medical record and Ward visit | Lexi-Comp® | 405 patients | NR | Evaluation of only pharmacokinetic DDIs; Study duration |
| Marusic et al., 2013 | Croatia | Prospective longitudinal | 3 months | Internal medicine | Medical record and Ward visit | Lexi-Comp® | 222 patients | NR | Patient follow-up time was short; Only one database used |
| De Paepe et al., 2013 [35] | Belgium | Cross-sectional | 0.75 month | Emergency | Medical record | Lexi-Comp® | 82 patients | 18 | Study duration; Underreporting of patient history |
| Bucşa et al., 2013 [37] | Romania | Prospective longitudinal | 3 months | Internal medicine | Medical record and Ward visit | Micromedex® | 305 patients | 14 | Faulty documentation and/or information; Monocentric study |
| Marino et al., 2016 [36] | Italy | Cross-sectional | 11 months | Emergency | Medical record | Micromedex® | 3,473 patients | 464 | Faulty documentation and/or information; Monocentric study |

NR—Not reported

Of the 6,540 patients included in this meta-analysis, 710 had clinically manifested DDIs. The pooled prevalence of clinically manifested DDIs was 9.2% (CI 95% 4.0–19.7). The lowest proportion of clinically manifested DDIs was found among patients attended in the emergency setting, followed by internal medicine (Table 2). Patients hospitalized in geriatric and intensive care units were more likely to have clinically manifested interactions during hospitalization (Fig 2).

One study in UCI subgroup included 400 participant patients, and proportion was 64.0% (CI 95%: 59.2–68.6) (Table 3). The mean number of medications per patient reported in six studies [32–35,37,40] ranged from 4.0 to 9.0, with an overall average of 5.47 ± 1.77 drugs per patient.

## Detection of drug interactions

To identify clinically manifested DDIs, medical records and ward visits (n = 7) [31–33,37–40] and medical records only (n = 3) [34–36] were used. The electronic databases used in the included studies were: Lexi-Comp® (n = 3) [31,32,35], Micromedex® (n = 3) [34,36,37], Stocley® (n = 1) [33], and Epocrates® (n = 1) [39]. Egger et al. (2003) [38] did not report the database used to identify DDIs (Table 1). None of the studies evaluated used more than one electronic database. In addition, five of the included studies reported that a pharmacist did not

**Table 2. Prevalence of drug interactions in hospitalized patients.**

| Author, year | Sample | Sample size | Average of number of drugs per patient | Prevalence of clinically manifested DDIs [%] (95% CI) |
|---|---|---|---|---|
| Herr et al., 1992 | Patients | 340 | NR | 1.5 (0.6–3.4) |
| Egger et al., 2003 | Patients | 163 | NR | 14.7 (10.1–21.0) |
| Blix et al., 2008 | Patients | 827 | 4.8 | 8.8 (7.1–11.0) |
| Fokter et al., 2009 | Patients | 323 | 5.0 | 1.2 (0.5–3.1) |
| Ray et al., 2010 | Patients | 400 | 9.0 | 64.0 (59.2–68.6) |
| Muñoz-Torrero et al., 2010 | Patients | 405 | 5.0 | 26.4 (22.4–30.9) |
| Marusic et al., 2013 | Patients | 222 | NR | 9.5 (6.3–14.0) |
| De Paepe et al., 2013 | Patients | 82 | 5.0 | 18.3 (11.4–28.0) |
| Bucşa et al., 2013 | Patients | 305 | 4.0 | 3.6 (2.0–6.4) |
| Marino et al., 2016 | Patients | 3473 | NR | 5.6 (4.9–6.4) |

NR—Not reported.

participate in the detection of drug interactions [32,34–36,40], and a pharmacist was a part of the team that evaluated the drug interactions in only three of the studies [33,37,38].

The severity of clinically manifested DDIs was reported in two studies [33,40]. In these studies, mildly manifested DDIs occurred in 1.36% (n = 127) of patients, moderate DDIs in 39.41% (n = 121), and severe DDIs in 15.96% (n = 49). Five different terminologies that address manifested DDIs were identified. Three studies [32–34] did not report the terminology used for clinically manifested DDIs, whereas only five studies [31,35,37,39,40] standardized the definition of terminologies used to refer to the manifested DDIs. The definitions and the terminologies used for manifested DDIs in these studies are described in Table 4.

### Assessment of methodological quality

About the results of the quality assessment, the control-case study was awarded 8/10 stars, which indicated a good methodological quality (S2 Table). Of the cross-sectional and prospective studies, two were of low quality [38,40], four were of reasonable quality [31,33,35,36] and three were of good quality [34,37,40] (S3 Table).

## Discussion

Although a significant proportion of inpatients are exposed to potential DDIs [21,29,33,35,37], approximately 1/10 of hospitalized patients had a clinically manifested DDI confirmed through laboratory testing, chart review and/or physical examination. In this scenario, strategies to prevent and resolve DDIs should not only be made from potential DDI information gathered from electronic bases [21,41,42]. The use of these databases by prescribers to generate alerts aimed at the prevention of clinically manifested DDIs may overestimate the problem and may lead to unnecessary interventions. In addition, these alerts may complicate the clinical workflow and lead to conflicts among health professionals [21,43,44].

This meta-analysis showed that the prevalence of clinically manifested DDIs in ICU patients (64.0%) is higher than among non-ICU inpatients. A previous systematic review observed that ICU patients have a higher prevalence of potential DDIs (67%) compared to non-ICU inpatients (33%) [21]. The lower prevalence of both potential DDIs and clinically manifested DDIs in non-ICU inpatients may be related to factors such as the decreased number of prescribed drugs as well as lower use of medicines with narrow therapeutic index when compared to UCI patients, and a lower rate of patients with organ failure [21,46].

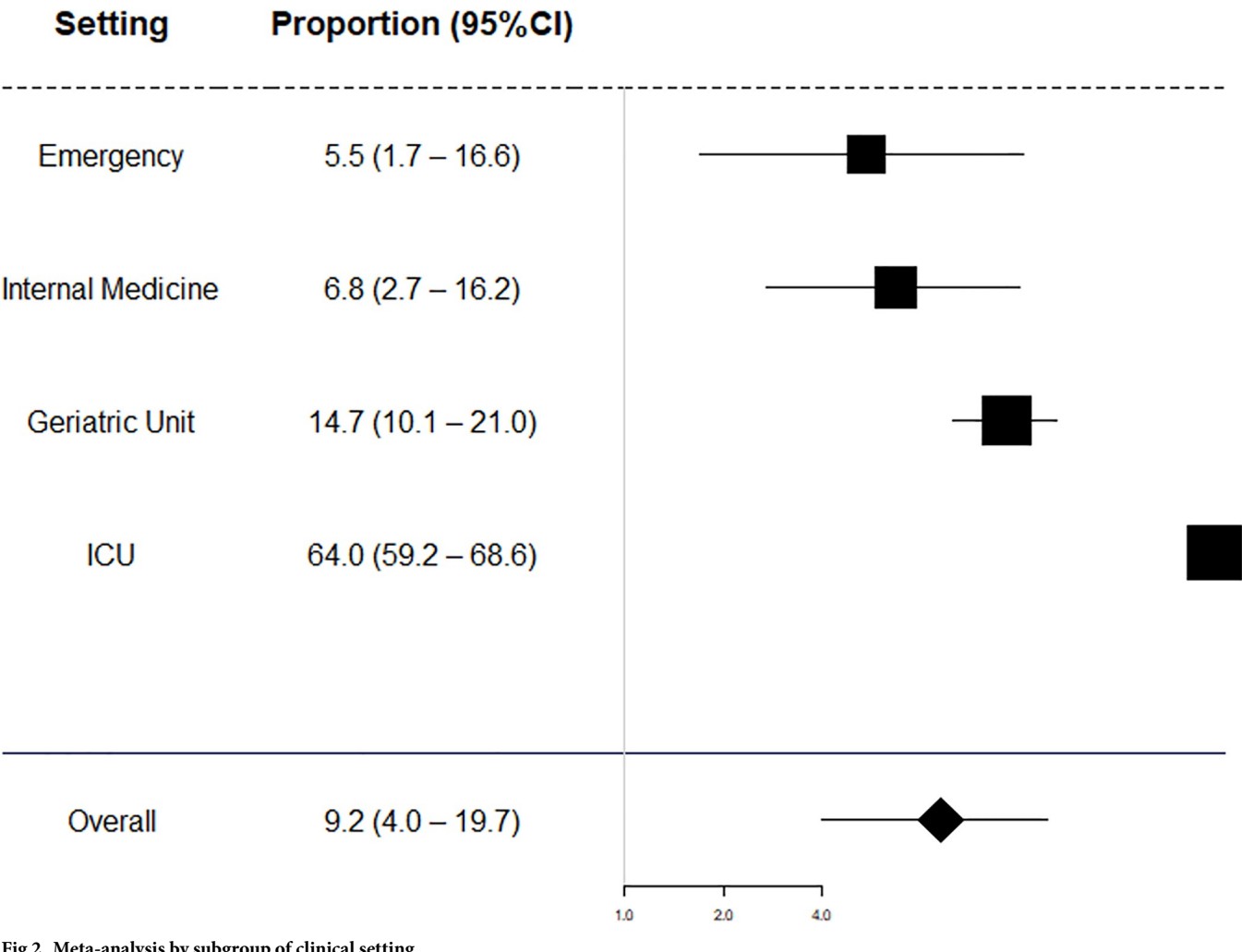

**Fig 2. Meta-analysis by subgroup of clinical setting.**

The best models of DDI prevention and management combine DDI warning systems with a pharmacist's assessment, thereby avoiding "alert fatigue" for DDIs that are not always clinically manifested [45]. According to Andrade (2015), a careful review of medical records can also be effectively used to detect DDIs in clinical practice [46]. This corroborates our findings, in which the review of medical records and interviews with patients were the most frequently

**Table 3. The overall proportion of clinically manifested DDIs according to practice setting.**

| Setting | Number of studies | Pooled proportion of clinically manifested DDIs (95% CI) | $I^2$ (%) |
|---|---|---|---|
| Emergency | 3 | 5.5 (1.7–16.6) | 94.5 |
| Internal Medicine | 5 | 6.8 (2.7–16.2) | 97.1 |
| Geriatric Unit | 1 | 14.7 (10.1–21.0) | - |
| ICU | 1 | 64.0 (59.2–68.6) | - |
| Overall | 10 | 9.2 (4.0–19.7) | 99 |

ICU—intensive care unit.

**Table 4. Terminologies used in the studies included in this review.**

| Reference | Terminology used | Definition of clinically manifested DDI |
|---|---|---|
| Herr et al., 1992 | Positive drug interaction | At least one sign indicated a drug interaction |
| Egger et al., 2003 | Clinically relevant drug interaction | NR |
| Blix et al., 2008 | NR | NR |
| Fokter et al., 2009 | NR | NR |
| Ray et al., 2010 | Adverse reaction caused by drug interaction | If drug interactions caused an adverse reaction |
| Muñoz-Torrero et al., 2010 | NR | NR |
| Marusic et al., 2013 | Actual drug–drug interactions | When a drug interaction causes an adverse drug reaction |
| De Paepe et al., 2013 | Clinically relevant drug interactions | When drug interactions caused drug withdrawal and/or dose modification |
| Bucşa et al., 2013 | Drug-drug interactions cause adverse drug reactions | A drug interaction that resulted in one or more adverse reactions |
| Marino et al., 2016 | Actual drug-drug interactions | NR |

NR—Not reported.

used methods and detected a greater number of manifested DDIs compared with other methods presented in this review.

Databases for DDIs are commonly used to help health professionals to prevent, identify, and resolve DDIs [47,48]. The differences and/or similarities between databases that are used to identify DDIs are related to the type of evidence (based on literature or spontaneous reports), the classification of DDI severity, the inclusion of medication doses for DDI assessment, the frequency in which each tool is updated, the sensitivity (the number of DDI pairs enrolled), and the specificity (a tool focused on a pharmacological class or type of patients) [48]. According to Hammar et al. (2015), researchers usually record all DDIs detected using an electronic database, without concern for clinical relevance [41]. Consequently, there is an overestimation in the identification of theoretically identified DDIs that does not reflect the reality of clinical practice. Recent studies have reported that increased sensitivity related to identification of clinically manifested DDIs may occur when two or more DDI-related research programs are combined [48–52]. Therefore, the use of only one database for the identification of DDIs in the included studies may justify the high prevalence of DDIs not clinically manifested in this study.

The degree of severity of DDIs is one of the most important criteria for clinical decision support [53]. According to Phansalkar (2013), the clinical information that provides context for DDIs is not readily available in electronic databases [17]. Therefore, the potential risk-benefit of DDIs requires the careful analysis of patient characteristics and diseases [54]. The present review revealed that severity ratings were not assessed in most studies; these results were in accordance with Roblek et al., 2015, who documented the low severity ratings of potential DDIs identified in 38 observational studies that evaluated the usability and adequacy of commercially available electronic databases that assess the prevalence of potential DDIs [47]. Thus, future studies should address the severity of DDIs and their association with the manifestations of signs and symptoms in patients. Nevertheless, the degree of severity does not influence the clinical manifestation of drug interactions.

In addition, the prevalence of manifested DDIs with lesser severity was higher than that of DDIs with greater severity, suggesting that the clinical relevance of DDIs should not be based solely on the degree of severity, as the probability of causing adverse outcomes is as important

as the severity of the outcome [17]. Therefore, the monitoring of specific cases of DDIs by health professionals is essential for the management of pharmacotherapy when necessary, and to minimize the deterioration of the patient's clinical condition.

To improve the quality of literature related to DDIs and to promote the comparison of the rates of prevalence of DDIs between studies, there should be no ambiguity in definitions and in research methods [21,22]. In the pharmacy domain, there is a lack of standardization of the terms and concepts of clinical practice [55,56]. This lack of uniformity between studies can generate confusion and lead to difficulties in the consolidating of this approach in clinical practice [22,56]. Thus, the terminologies and concepts for manifested DDIs in the studies showed heterogeneity, which hinders the development of an ideal definition to refer to manifested DDIs. In addition, some studies did not present clear information on the methods used for the identification of clinically manifested DDIs [33–36,38,39]. For example, in the USA, Hines et al. (2011) evaluated and discussed the problems associated with evidence databases for DDIs and revealed a lack of standardized terminologies and concepts or clear information on methods [57]. Consequently, it is necessary to discuss and to adopt terminologies, standardized concepts, and methods to detect clinically manifested DDIs, to compare the results obtained in the studies, and to optimize the methods of prevention, identification, and management of DDIs.

On assessment, the quality of majority of the included studies was moderate or good. Similar findings were observed in Dechanont's meta-analysis (2014), in which the quality of 13 cross-sectional studies upon admission associated with DDI was assessed [22]. In this context, there is no consensus on the best tool for quality assessment. In addition, quality assessment is influenced by subjective judgment and a lack of information on the studies [58].

Recently, Zheng et al. (2018) published a systematic review and meta-analysis on the harmful effects of DDIs among hospitalized patients. The present systematic review, and that of Zheng and his collaborators, have similar subjects and rationales: high volumes of DDI alerts lead to alert fatigue, in which prescribers ignore relevant DDI alerts when exposed to an excessive number of notifications. However, the present review is different from the review published in 2018 in many ways. First, although Zheng et al. (2018) included studies that reported the prevalence of DDIs in an inpatient setting, our review only included studies that reported DDIs confirmed by laboratory tests and/or by signs and symptoms documented in the medical records after analysis by specialists. Second, our literature search included more databases, data were extracted from research conducted up to 2018, and Spanish-language publications were included [21]. Third, we included 10 studies in which data related to the prevalence of clinically manifested DDIs were fully available; nine of these were not included in the previous systematic review [21] Fourth, we obtained different results and the present systematic review observed that 1/10 inpatients experienced at least one clinically manifested DDI. To the best of our knowledge, this is the first review to identify the terminologies and concepts for clinically manifested DDIs used in the included studies.

Nonetheless, our study also has some limitations: most of the investigated studies had some flaws related to sample size that may interfere with the prevalence rate and statistical heterogeneity was observed across studies ($I^2$ was greater than 95% in one setting subgroup and it could not be obtained in two subgroups). In addition, although the authors of the included studies stated that clinical manifestations were suspected to be a result of DDIs, a potential bias in the assessment of causality of clinical manifestations should not be overlooked.

## Conclusion

This systematic review showed that, despite the significant prevalence of potential DDIs reported in the literature, less than one in ten patients were exposed to a clinically manifested

drug interaction. However, UCI patients were considerably more likely to experience these adverse events than non-UCI patients. Once clinically manifested drug interactions are associated with the length of hospital stay, the early detection and resolution of this events are paramount, especially in times of high ICU bed occupancy rates.

In addition, an understanding of the prevalence of the clinical manifestation of DDIs in patients can optimize the work process of several health professionals in the hospital environment, as it reduces the incidence of alert fatigue, enhances decision-making for DDI prevention or resolution, and, consequently, contributes to patient safety.

In view of these results, the authors suggest the use of causality tools to identify clinically manifested DDIs as well as clinical adoption of DDI lists based on actual adverse outcomes that can be identified through the implementation of real DDI notification systems. Moreover, the lack of standardized terminology and definitions can generate confusion and difficulty in the resolution of clinical manifestations caused by DDIs. The use of more than one electronic database combined with the analysis of medical records and ward visits by health professionals may contribute to more accurate identification of clinically manifested DDIs.

Future studies employing a prospective design would be more suitable for the identification and the resolution of clinical manifestations caused by drug interactions in hospitalized patients. Finally, further studies should focus on risk factors for patients with clinically manifested DDIs, to help practicing clinicians and pharmacists to identify at risk patients.

## Supporting information

**S1 Checklist. PRISMA 2009 checklist.**
(DOC)

**S1 Table. Complete search strategy in the searched databases.**
(DOCX)

**S2 Table. Quality score of case-control studies.** *Studies that scored > 5 stars were considered of good quality.
(DOCX)

**S3 Table. Quality Assessment Tool for Observational Cohort and Cross-Sectional Studies.**
(DOCX)

## Author Contributions

**Conceptualization:** Tâmara Natasha Gonzaga de Andrade Santos, Divaldo Pereira de Lyra, Jr., Alfredo Dias de Oliveira Filho.

**Formal analysis:** Tâmara Natasha Gonzaga de Andrade Santos, Givalda Mendonça da Cruz Macieira, Bárbara Manuella Cardoso Sodré Alves, Geovanna Cunha Cardoso, Mônica Thaís Ferreira Nascimento, Paulo Ricardo Saquete Martins-Filho.

**Investigation:** Tâmara Natasha Gonzaga de Andrade Santos, Givalda Mendonça da Cruz Macieira, Bárbara Manuella Cardoso Sodré Alves, Thelma Onozato, Geovanna Cunha Cardoso, Paulo Ricardo Saquete Martins-Filho, Divaldo Pereira de Lyra, Jr., Alfredo Dias de Oliveira Filho.

**Methodology:** Tâmara Natasha Gonzaga de Andrade Santos, Givalda Mendonça da Cruz Macieira, Bárbara Manuella Cardoso Sodré Alves, Thelma Onozato, Geovanna Cunha Cardoso, Paulo Ricardo Saquete Martins-Filho, Divaldo Pereira de Lyra, Jr., Alfredo Dias de Oliveira Filho.

**Supervision:** Alfredo Dias de Oliveira Filho.

**Writing – original draft:** Tâmara Natasha Gonzaga de Andrade Santos, Givalda Mendonça da Cruz Macieira, Bárbara Manuella Cardoso Sodré Alves, Thelma Onozato, Mônica Thaís Ferreira Nascimento, Paulo Ricardo Saquete Martins-Filho, Divaldo Pereira de Lyra, Jr., Alfredo Dias de Oliveira Filho.

**Writing – review & editing:** Tâmara Natasha Gonzaga de Andrade Santos, Givalda Mendonça da Cruz Macieira, Bárbara Manuella Cardoso Sodré Alves, Thelma Onozato, Mônica Thaís Ferreira Nascimento, Paulo Ricardo Saquete Martins-Filho, Divaldo Pereira de Lyra, Jr., Alfredo Dias de Oliveira Filho.

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
