## [Decision Letter · Decision Letter 0]

28 Apr 2020

PONE-D-19-34442

Prevalence of clinically manifested drug interactions in hospitalized patients: a systematic review and meta-analysis

PLOS ONE

Dear Mr Oliveira Filho,

Thank you for submitting your manuscript to PLOS ONE. After careful consideration, we feel that it has merit but does not fully meet PLOS ONE’s publication criteria as it currently stands. Therefore, we invite you to submit a revised version of the manuscript that addresses the points raised during the review process.

We would appreciate receiving your revised manuscript by Jun 11 2020 11:59PM. To enhance the reproducibility of your results, we recommend that if applicable you deposit your laboratory protocols in protocols.io, where a protocol can be assigned its own identifier (DOI) such that it can be cited independently in the future. For instructions see: http://journals.plos.org/plosone/s/submission-guidelines#loc-laboratory-protocols

We look forward to receiving your revised manuscript.

Kind regards,

Jed N. Lampe, Ph.D.

Academic Editor

PLOS ONE

Journal Requirements:

3. Please kindly add your tables to be part of the manuscript and remove the uploaded table files.

4. Please consider including Forest plot representation of your data.

6. Thank you for stating the following towards the end of your manuscript:

'Financial Support

This study was financed in part by the Coordenação de Aperfeiçoamento de

 Pessoal de Nível Superior - Brasil (CAPES) - Finance Code 001. The funder of the study had no role in study design, data collection, data 435 analysis, data interpretation, or writing of the report.'

'The author(s) received no specific funding for this work.'

7. Please include captions for your Supporting Information files at the end of your manuscript, and update any in-text citations to match accordingly. Please see our Supporting Information guidelines for more information: http://journals.plos.org/plosone/s/supporting-information

Additional Editor Comments:

Please adhere strictly to all PLOS One publications requirements.

Reviewers' comments:

Reviewer's Responses to Questions

**Comments to the Author**

1. Is the manuscript technically sound, and do the data support the conclusions?

Reviewer #1: Yes

2. Has the statistical analysis been performed appropriately and rigorously? 

Reviewer #1: Yes

3. Have the authors made all data underlying the findings in their manuscript fully available?

Reviewer #1: Yes

4. Is the manuscript presented in an intelligible fashion and written in standard English?

Reviewer #1: Yes

5. Review Comments to the Author

Reviewer #1: Is the manuscript technically sound, and do the data support the conclusions?

in the method part exactly define the PICO (patient, intervention, comparator and outcome) of your study

Has the statistical analysis been performed appropriately and rigorously?

determine the forest plot for your study

Have the authors made all data underlying the findings in their manuscript fully available?

yes

delete the references from the abstract

mention the limitation and bias of your study

6. PLOS authors have the option to publish the peer review history of their article (what does this mean?). If published, this will include your full peer review and any attached files.

Reviewer #1: No

---

## [Author Response · Author response to Decision Letter 0]

11 Jun 2020

Reviewer # Academic Editor:

A.: Modifications were made in accordance with the reviewer’s suggestion.

2) Please ensure that you include a title page within your main document. We do appreciate that you have a title page document uploaded as a separate file, however, as per our author guidelines (http://journals.plos.org/plosone/s/submission-guidelines#loc-title-page) we do require this to be part of the manuscript file itself and not uploaded separately. 

A.: Page 01 – Ok. Modifications were made in accordance with the reviewer’s suggestion.

3) Please kindly add your tables to be part of the manuscript and remove the uploaded table files.

A.: Page 09 - 14 - Ok. It was corrected.

4. Please consider including Forest plot representation of your data.

A.: We agree. The figure was inserted into the system during submission.

5) PLOS requires an ORCID iD for the corresponding author in Editorial Manager on papers submitted after December 6th, 2016. Please ensure that you have an ORCID iD and that it is validated in Editorial Manager. To do this, go to ‘Update my Information’ (in the upper left-hand corner of the main menu), and click on the Fetch/Validate link next to the ORCID field. This will take you to the ORCID site and allow you to create a new iD or authenticate a pre-existing iD in Editorial Manager. Please see the following video for instructions on linking an ORCID iD to your Editorial Manager account: https://www.youtube.com/watch?v=_xcclfuvtxQ

A.: Ok. was inserted into the system during submission.

6) Thank you for stating the following towards the end of your manuscript:

'Financial Support

This study was financed in part by the Coordenação de Aperfeiçoamento de Pessoal de Nível Superior - Brasil (CAPES) - Finance Code 001. The funder of the study had no role in study design, data collection, data 435 analysis, data interpretation, or writing of the report.'

'The author(s) received no specific funding for this work.'

Ok, it was corrected. We included the amended statements within our cover letter.

Please clarify the sources of funding (financial or material support) for your study. List the grants or organizations that supported your study, including funding received from your institution.

State what role the funders took in the study. If the funders had no role in your study, please state: “The funders had no role in study design, data collection and analysis, decision to publish, or preparation of the manuscript.”

If any authors received a salary from any of your funders, please state which authors and which funders.

If you did not receive any funding for this study, please state: “The authors received no specific funding for this work.”

A.: Thank you for all considerations. The text about Financing Statement has been removed from the manuscript. Additional, this text has been added in the cover letter and in the online submission form.

7) Please include captions for your Supporting Information files at the end of your manuscript, and update any in-text citations to match accordingly. Please see our Supporting Information guidelines for more information: http://journals.plos.org/plosone/s/supporting-information.

A.: Page 27- Modifications were made in accordance with the reviewer’s suggestion.

Reviewer #1

1) Is the manuscript technically sound, and do the data support the conclusions?

A.: Lines 349-352 – Thank you. Modifications were made in accordance with the reviewer’s suggestion.

in the method part exactly define the PICO (patient, intervention, comparator and outcome) of your study

A.: Lines 102-106 - Modifications were made in accordance with the reviewer’s suggestion.

Has the statistical analysis been performed appropriately and rigorously?

determine the forest plot for your study

A.: We agree. The figure was inserted into the system during submission.

Delete the references from the abstract

A.: Line 42- We're really sorry and thank you for the suggestion. It's fixed now..

Mention the limitation and bias of your study

A.: Lines 339-341 - Modifications were made in accordance with the reviewer’s suggestion.

---

## [Editor Report · Decision Letter 1]

15 Jun 2020

Prevalence of clinically manifested drug interactions in hospitalized patients: a systematic review and meta-analysis

PONE-D-19-34442R1

Dear Dr. Oliveira Filho,

We’re pleased to inform you that your manuscript has been judged scientifically suitable for publication and will be formally accepted for publication once it meets all outstanding technical requirements.

Kind regards,

Jed N. Lampe, Ph.D.

Academic Editor

PLOS ONE

---

## [Editor Report · Acceptance letter]

18 Jun 2020

PONE-D-19-34442R1 

Prevalence of clinically manifested drug interactions in hospitalized patients: a systematic review and meta-analysis 

Dear Dr. Oliveira Filho:

I'm pleased to inform you that your manuscript has been deemed suitable for publication in PLOS ONE. Congratulations! Your manuscript is now with our production department. 

Kind regards, 

on behalf of

Dr. Jed N. Lampe 

Academic Editor

PLOS ONE